# Prophylactic Hyperthermic Intraperitoneal Chemotherapy (HIPEC) for Gastric Cancer—A Systematic Review

**DOI:** 10.3390/jcm8101685

**Published:** 2019-10-15

**Authors:** H. J. F. Brenkman, M. Päeva, R. van Hillegersberg, J. P. Ruurda, N. Haj Mohammad

**Affiliations:** 1Department of Surgery, University Medical Center Utrecht, Utrecht University, 3584 CX Utrecht, The Netherlands; h.j.f.brenkman@umcutrecht.nl (H.J.F.B.); r.vanhillegersberg@umcutrecht.nl (R.v.H.); j.p.ruurda@umcutrecht.nl (J.P.R.); 2Department of Medical Oncology, University Medical Center Utrecht, Utrecht University, 3584 CX Utrecht, The Netherlands; mariliispaeva@hotmail.com

**Keywords:** stomach neoplasms, hyperthermia (induced), peritoneal infusions

## Abstract

Survival after potentially curative treatment of gastric cancer remains low, mostly due to peritoneal recurrence. This descriptive review gives an overview of available comparative studies concerning prophylactic hyperthermic intraperitoneal chemotherapy (HIPEC) for patients with gastric cancer with neither clinically evident metastases nor positive peritoneal cytology who undergo potentially curative gastrectomy. After searching the PubMed, Embase, CDSR, CENTRAL and ASCO meeting library, a total of 11 studies were included comparing surgery plus prophylactic HIPEC versus surgery alone (SA): three randomised controlled trials and eight non-randomised comparative studies, involving 1145 patients. Risk of bias was high in most of the studies. Morbidity after prophylactic HIPEC was 17–60% compared to 25–43% after SA. Overall survival was 32–35 months after prophylactic HIPEC and 22–28 months after SA. The 5-year survival rates were 39–87% after prophylactic HIPEC and 17–61% after SA, which was statistically significant in three studies. Peritoneal recurrence occurred in 7–27% in the HIPEC group, compared to 14–45% after SA. This review tends to demonstrate that prophylactic HIPEC for gastric cancer can be performed safely, may prevent peritoneal recurrence and may prolong survival. However, studies were heterogeneous and outdated, which emphasizes the need for well-designed trials conducted according to current standards.

## 1. Introduction

Gastric cancer is the fifth most common malignancy in the world and the third leading cause of cancer death [1]. The median survival of patients undergoing treatment with curative intent, including perioperative chemotherapy and gastrectomy, is approximately 50 months [2]. The most important reason for treatment failure is peritoneal recurrence, which develops in up to 70% of the cases and has an average survival of merely four months after diagnosis [3].

In patients with peritoneal recurrence, palliative chemotherapy is standard of care in order to try to eradicate cancer cells, but peritoneal penetration of chemotherapy is poor. To overcome this limitation, hyperthermic intraperitoneal chemotherapy (HIPEC) has been introduced [4,5]. The combination of HIPEC with cytoreductive surgery is the standard treatment for several peritoneal malignancies including colorectal cancer, pseudomyxomas and mesotheliomas [6,7,8], and has also been used as a therapeutic option for peritoneal metastases from gastric cancer [9,10].

Alternatively, prevention of peritoneal recurrence might be a better method to improve prognosis after curative treatment. For this reason, prophylactic HIPEC has been suggested as an adjuvant treatment strategy in patients with a high risk for peritoneal recurrence after potentially curative treatment [9,10]. This review was conducted in order to summarize the available comparative studies concerning prophylactic HIPEC for gastric cancer patients without proven metastatic disease who undergo resection with curative intent.

## 2. Materials and Methods

### 2.1. Literature Search Strategy

An electronic literature search was conducted using the databases of PubMed, Embase, Cochrane Database of Systematic Reviews, Cochrane Central Register of Controlled Trials, American Society of Clinical Oncology (ASCO) meeting library. The period for the search was from 1980 to June 2019. The search terms included ‘prophylactic’, ‘hyperthermic’, ‘intraperitoneal’, ‘chemotherapy’, ‘gastric cancer’ and their synonyms in various combinations. The search also included all MeSH terms. The reference lists of extracted articles were reviewed for further identification of relevant studies. The titles and abstracts were inspected independently by two authors (M.P. and N.H.M.).

### 2.2. Study Selection Criteria

Study type: Comparative studies.Study characteristics: Only articles written in English or Dutch were included. Papers for which no abstract or no full text was available were excluded. In case of multiple publications of a study, only the most recent version was included.Participants: Patients with primary cancer of the stomach without peritoneal or distant metastases who underwent radical resection were included. Peritoneal cytology positive for cancer cells is regarded as a proven metastatic disease in TNM-7, thus patients with positive peritoneal cytology were excluded [11].Intervention and comparison: Patients who underwent radical surgery in combination with prophylactic HIPEC formed the intervention group. The comparison group consisted of patients receiving surgery alone (SA). No selection was made based on the lymphadenectomy performed.Outcomes: The primary endpoint was overall survival. The secondary endpoints were 5-year survival, disease-free survival, peritoneal recurrence, post-operative morbidity and mortality and quality of life.

### 2.3. Assessment of Methodological Quality

Assessment of the risk of bias of the selected studies was conducted independently by two authors (M.P., H.B.). Any disagreements between the authors were resolved by consulting a third author (N.H.M.). For the evaluation of randomised controlled trials, the Risk of Bias Tool 2.0 provided by Cochrane was implemented [12]. ROBINS-I by Cochrane was used for the non-randomised comparative studies [13]. 

## 3. Results

### 3.1. Literature Search

Database search delivered 545 articles (Figure 1). After screening the titles and abstracts, the full texts of 63 articles were assessed for eligibility. In addition to the electronic search, four articles were retrieved after the citation screening, and one abstract was included from ASCO. Finally, ten articles fulfilled the criteria, which included 11 studies: three randomised controlled trials and eight non-randomised comparative studies. The studies reported most of the selected endpoints, except for quality of life, which was not measured in any of the studies.

### 3.2. Assessment of Risk of Bias

Table 1 and Table 2 demonstrate the results of the risk of bias assessment. Most of the studies showed a high risk of bias. The studies of Murata et al. and Yonemura et al. (2001) were only available as an abstract, which is why they could not be assessed.

### 3.3. Treatment

Most of the studies were conducted in Asia with only one study having been performed in Europe (Table 3 and Table 4). There were ten studies that performed HIPEC for T3-4 tumors: seven of these studies included patients with clinically diagnosed T3-4 tumors, and three included pathologically confirmed T3-4 tumors. Mitomycin C was used in the therapy regimen in ten studies, whether alone, in combination with cisplatin, etoposide, both cisplatin and etoposide or cisplatin with 5-FU. In one study, only cisplatin in combination with paclitaxel was used. Temperature during HIPEC ranged from 40 to 45 degrees, and the duration of the procedure ranged from 30 to 120 min. Studies performed either a D1 or D2 lymphadenectomy during gastrectomy, which did not differ between treatment arms.

### 3.4. Morbidity and Mortality

Data on morbidity were provided in two randomised controlled trials [14,23] and four non-randomised comparative studies [14,17,19,21]. Total numbers for morbidity ranged from 16.7% to 60% after prophylactic HIPEC and from 25% to 42.5% after SA.

When types of complications were viewed separately, a significantly higher occurrence of respiratory failure was found after prophylactic HIPEC in the non-randomised controlled study by Kunisaki et al. (73% vs. 19%, *p* < 0.0001) [19]. Other studies did not show a difference in respiratory failure. Anastomotic leak was reported in two randomised controlled trials [14,23], and three non-randomised controlled studies [14,17,19], with an incidence rate ranging from 2% to 20% in the HIPEC group versus 3% to 15% in the SA group. The anastomotic leakage rates did not differ between the two groups in any of the studies. Ileus was reported in one randomised controlled trial [14] and two non-randomised controlled studies [14,19], with the occurrence rate ranging from 2% to 4% in the HIPEC arm compared with 4% to 7.1% in the SA arm. Pancreatic fistula was recorded in two non-randomised comparative studies, with an incidence rate ranging from 20% to 39% in the intervention group as opposed to 7.5% to 46% in the control group [17,19]. Bowel fistula and bone marrow suppression were recorded in one randomised controlled trial, in which they both occurred once (2.1%) in the intervention group [23]. Other reported complications included renal failure, intra-abdominal abscess, liver dysfunction, bleeding and biliary fistula.

Mortality was described in one randomised controlled trial [15] and three non-randomised controlled studies [14,16,17], ranging from 0% to 16.7% after prophylactic HIPEC versus 0% to 9.8% after SA. There were no significant differences between the two groups.

### 3.5. Survival

The median follow up of the studies ranged from 14.6 months to five years.

Overall survival was reported in one randomised controlled trial [23] and three non-randomised comparative studies [17,20,21], ranging from 32 to 34.6 months in the intervention group and from 22 to 28.2 months in the control group. Although overall survival was higher in the HIPEC group compared to the SA group in all studies, this was significant in only one non-randomised study (Hirose et al., 33 vs. 22 months, *p* = 0.0142) [17]. 

Five-year survival was reported in two randomised controlled trials [21,23] and six non-randomised comparative studies [14,16,17,18,19,22], ranging from 39.1% to 86.8% after prophylactic HIPEC and 17.3% to 61% after SA. The 5-year survival was higher in the HIPEC group compared to the SA group in two randomised controlled trials although no statistical significance was reported [19,23]. In the non-randomised comparative studies, the 5-year survival was higher in 5 of the 6 studies, being significantly higher in three studies [14,17,18]. 

### 3.6. Recurrence

Disease-free survival was reported in one non-randomised comparative study, where it showed a non-significant difference (34.5 months in the HIPEC group vs. 24.7 months in the SA group, *p*-value not reported) [21].

Data on peritoneal recurrence was published in two randomised controlled trials [13,22] and four non-randomised controlled studies [14,17,19,22], ranging from 6.8% to 26.7% in the HIPEC arm and from 14.1% to 45% in the SA arm. All studies showed a trend towards a lower peritoneal recurrence rate after prophylactic HIPEC. However, only one non-randomised study showed a significant difference (HR = 0.20 (95% CI 0.068–0.61), *p* = 0.005) [18].

## 4. Discussion

Peritoneal recurrence is the most important reason for treatment failure after gastrectomy with curative intent. This descriptive review demonstrates that prophylactic HIPEC may extend survival and prevent peritoneal recurrence in patients with a T3-4 tumor without metastases or positive peritoneal cytology. Prophylactic HIPEC was performed during gastrectomy in all studies and did not significantly raise the morbidity or mortality rate.

The results on survival outcomes and peritoneal recurrence coincide with previously published meta-analyses on prophylactic HIPEC for gastric cancer [24,25,26,27,28]. Sun et al. observed a significant improvement in the overall survival in the HIPEC group (RR 0.73), as did Yan et al. (HR 0.60) [25,27]. Sun et al. found prophylactic HIPEC to lead to a lower recurrence rate compared to the control group (RR 0.45) [25]. Nevertheless, these meta-analyses differ from this review as they included patients with positive peritoneal cytology as well. The current review excluded these patients, as positive cytology is recognised as a predictor of peritoneal recurrence and is classified as a proven metastatic disease in the TNM-7 [11,29]. Performing HIPEC in patients with positive cytology might therefore be considered as therapeutic HIPEC rather than prophylactic HIPEC. This may lead to lower survival rates when compared to studies including only prophylactic HIPEC.

Results on morbidity and mortality rates are conflicting in previous studies. According to Huang et al., application of intraperitoneal chemotherapy increases the risk of bone marrow depression (OR = 5.74), fever (OR = 3.67) and intra-abdominal abscess (OR = 3.57) compared to surgery alone [26]. Conflictingly, Mi et al. found prophylactic HIPEC not to be associated with higher risks of myelosuppresion, anastomotic leakage, ileus or bowel perforation compared to surgery alone [24]. According to this review, morbidity and mortality are comparable between the groups with only respiratory failure occurring significantly more often after prophylactic HIPEC. 

There are several limitations to this review. The total number of patients included in this paper is small, and the information is often extracted from subgroup data. Moreover, the majority of the studies have a high risk of bias. Most of the studies were published more than 10 years ago. Since publication of the MAGIC-trial in 2006, perioperative chemotherapy has been standard of care in Europe [30]. Recently, the FLOT4 study established FLOT as the chemotherapy regimen of choice [2]. However, most of the included studies did not administer perioperative chemotherapy, which may have resulted in lower efficacy. Furthermore, new techniques as positron emission tomography and diagnostic laparoscopy have improved staging and therefore may have led to better patient selection. Moreover, gastric cancer care in general has changed including the shift from laparotomy to minimally invasive techniques, and techniques of HIPEC have changed as well. Most of the studies executed HIPEC intraoperatively after the reconstructions, but, since the late 1990s, intraperitoneal chemotherapy is also executed before completing the anastomoses. It is suggested that this method may lead to less adverse effects and potentially decreases the risk of locoregional recurrence at the anastomotic site; however, there is no data to support this hypothesis [22,31]. Another option which has been used in prophylactic HIPEC for colorectal cancer and may be used for gastric cancer in the future as well is to perform HIPEC several weeks after surgery, which is also suggested to reduce surgical complications [32]. There is considerable heterogeneity between the studies, as patient and tumor characteristics differed between the studies. In addition, the technique, temperature, time and chemotherapeutic agent of HIPEC varied between the studies. This heterogeneity between the studies may also explain the conflicting results reported in the studies, for instance on morbidity and mortality. Finally, the results gathered in Asian studies cannot be fully applied to Western populations, as survival rates and tumor characteristics of gastric cancer differ greatly between Asian and non-Asian populations being significantly favourable to Asian patients [33].

This review demonstrates the need for further well-designed prospective randomized studies comparing patients with gastric cancer without metastases who receive prophylactic HIPEC with patients receiving surgery only. In 2014, GASTRICHIP-trial was commenced, a prospective randomised phase III study conducted in France to evaluate the effects of prophylactic HIPEC on patients with gastric cancer involving the serosa and/or lymph node involvement and/or with positive cytology [34]. This currently on-going study is one of the first phase III trials conducted in a Western country that evaluates the effect of prophylactic HIPEC on gastric cancer. This study also includes translational research, which will be valuable for future studies. 

In a study monitoring quality of life (QoL) before and after HIPEC for peritoneal metastases from various origins, a reduction in QoL was seen that recovered in six months after the intervention [35]. Considering the relatively low life expectancy of patients diagnosed with gastric cancer, QoL must be investigated in the future studies along with efficacy and morbidity.

## 5. Conclusions

Prophylactic HIPEC for gastric cancer can be performed safely, and may prevent peritoneal recurrence, and may prolong survival of patients with neither clinically evident metastases nor positive peritoneal cytology. However, the heterogeneity and age of the studies in this review show the need for well-designed trials conducted according to current standards.

## Figures and Tables

**Figure 1 jcm-08-01685-f001:**
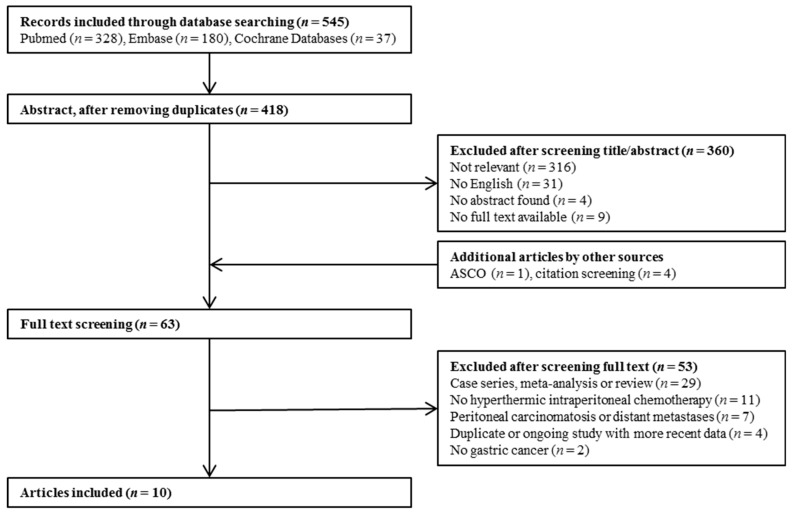
Article selection flowchart.

**Table 1 jcm-08-01685-t001:** Risk of bias of randomised controlled trials.

Author	Pub. Year	Total Number of Participants	Randomisation	Deviations from Intended Interventions	Missing Outcome Data	Measurement of Outcomes	Reported Result	Overall Risk of Bias
Koga et al. [14]	1988	60	Some concerns	High	Low	Low	Low	High
Kaibara et al. [15]	1989	82	Some concerns	Low	Low	Low	Low	Some concerns

**Table 2 jcm-08-01685-t002:** Risk of bias of non-randomised comparative studies.

Author	Pub. Year	Total No. of Participants	Confounding	Selection of Participants	Classification of Interventions	Deviations fromIntended Interventions	Missing Outcome Data	Measurement of Outcomes	Reported Result	Overall Risk of Bias
Koga et al. [14]	1988	137	Serious	Serious	Low	Low	Low	Moderate	Low	Serious
Yonemura et al. [16]	1995	160	Serious	Low	Low	Low	Low	Moderate	Moderate	Serious
Hirose et al. [17]	1999	55	Serious	Low	Low	Low	Low	Low	Moderate	Serious
Kim et al. [18]	2001	65	Serious	Moderate	Low	Low	Low	Moderate	Low	Serious
Kunisaki et al. [19]	2002	124	Serious	Low	Low	Low	Low	Moderate	Low	Serious
Kunisaki et al. [20]	2005	61	Serious	Low	Low	Low	Low	Moderate	Low	Serious
Coccolini et al. [21]	2016	34	Critical	Low	Low	Low	Low	Moderate	Low	Critical
Murata et al. [22]	2016	186	NI	NI	NI	NI	NI	NI	NI	NI

**Table 3 jcm-08-01685-t003:** Randomised controlled trials.

Author	Year	Country	Tumor Characteristics		No. of Participants		Therapy Regimen		Overall Survival		5-Year Survival		Peritoneal Recurrence		Morbidity and Mortality	
			HIPEC	SA	HIPEC	SA	HIPEC	SA	HIPEC	SA	HIPEC	SA	HIPEC	SA	HIPEC	SA
Koga et al. [14]	1988	Japan	cT3-4,	cT3-4	32	28	CHPP: MMC 8–10 mg/L, total dose 64–100 mg. Temperature: in 44–45 °C, out 40–42 °C. Time: 50–60 min. No CTx.	Surgery alone	NA	NA	NA	NA	NA	NA	Morbidity: NA Mortality: 3.1%	Morbidity: NA Mortality: 0%
Kaibara et al. [15]	1989	Japan	cT3-4	cT3-4	42	40	CHPP: MMC 10 mg/L, total dose 20 mg. Temperature: in 44–45 °C, out 40–42 °C. Time: 50–60 min. No CTx.	Surgery alone	NA	NA	71.5%	59.7%	11.9%	20%	Morbidity: 0%	Morbidity: 0%
Yonemura et al. [23]	2001	Japan	T3-4	T3-4	48	47	CHPP: MMC 30 mg + cisplatin 300 mg. Temperature: 42–43.5 °C. Time: 60 min. No CTx.	Surgery alone	HR 0.42 (95% CI 0.20–0.90)		61%	42%	12.5%	14.9%	Morbidity: NA	Morbidity: NA Mortality: 4.3%

NA = Not available; CHPP = Continuous Hyperthermic Peritoneal Perfusion; MMC = Mitomycin C; CTx = Chemotherapy; SA = Surgery Alone.

**Table 4 jcm-08-01685-t004:** Non-randomised comparative studies.

Author	Year	Country	Tumor Characteristics		No. of Participants		Therapy Regimen		Median Disease Free Survival		Overall Survival		5-Year Survival		Morbidity and Mortality		Peritoneal Recurrence	
			HIPEC	SA	HIPEC	SA	HIPEC	SA	HIPEC	SA	HIPEC	SA	HIPEC	SA	HIPEC	SA	HIPEC	SA
Koga et al. [14]	1988	Japan	cT3-4	cT3-4	59	78	CHPP: MMC 8–10 mg/L, total dose 64–100 mg. Temperature: in 44–45 °C, out 40–42 °C. Time: 50–60 min.	Surgery alone	NA	NA	NA	NA	63.0%	43.0%	Mortality: 15.6% Morbidity: NA	Mortality: 9.8% Morbidity: NA	6.8% (4/59) in 4 years	14.1% (1/78) in 4 years
Yonemura et al. [16]	1995	Japan	cT3-T4	cT3-T4	79	81	CHPP: 30 mg MMC + 300 mg CDDP. Temperature: 41.5–43.5 °C. Time: 60 min. Adj. CTx.: 2–3 weeks 400 mg UFT p/o.	Surgery + Adj. CTx.: 2–3 weeks 400 mg UFT p/o.	NA	NA	NA	NA	NA	NA	Mortality: 3.8%	Mortality: 2.5%	NA	NA
Hirose et al. [17]	1999	Japan	cT3-4	cT3-4	15	40	CHPP: CDDP 100 mg + MMC 20 mg + etoposide 100 mg. Temperature: 41–44.5 °C. Time: 50 min. Adj. CTx: MMC and 5FU	Surgery + Adj. CTx.: MMC and 5FU	NA	NA	33 m	22 m	39.1%	17.3%	Mortality: 0%. Morbidity: 60%	Mortality: 5%. Morbidity: 42,5%	26.7%	45%
Kim et al. [18]	2001	Korea	pT3-4	pT3-4	29	36	Before closure: MMC 10 µg/mL, 40 mg in total. Temperature: 42–44 °C. Time: 120 min. Adj. CTx.: >6 cycles 5-FU or 5-FU + MMC	Surgery + Adj. CTx.: >6 cycles 5-FU or 5-FU + MMC	NA	NA	NA	NA	58.5%	44.4%			NA	NA
Kunisaki et al. [19]	2002	Japan	T3-4	T3-4	45	79	CHPP: 15 mg MMC + 150 mg CDDP + 150 mg etoposide. Temperature.: 42–43 °C. Time: 40 min. Adj. CTx.	Surgery + Some patients adj. CTx.	NA	NA	NA	NA	48.5%	55.1%	Morbidity: NA	Morbidity: NA Mortality: 1.3%	24.4%	26.6%
Kunisaki et al. [20]	2005	Japan	cT3-4, linitis plastica	pT2-4, linitis plastica	6	55	CHPP: 15 mg MMC + 150 mg CDDP + 150 mg etoposide. Temperature.: 42–43 °C. Time: 40 min. Adj. CTx.	Surgery + Some patients adj. CTx.	NA	NA	32 m	24 m	NA	NA	NA	NA	NA	NA
Coccolini et al. [21]	2016	Italy	pT4	pT3-4	6	28	Neoadj. CTx. Before reconstruction open HIPEC: CDDP 100 mg/m2 + paclitaxel 75 mg/m^2^. Temperature: 40–41 °C. Time: 90 min	Neoadj. CTx. + Surgery	34.5 m	24.7 m	34.6 m	27.7 m	NA	NA	Morbidity: 16.7%	Morbidity: 16.4%	NA	NA
Murata et al. [22]	2016	Japan	pT3-4	pT3-4	186 in total		MMC + CDDP +/− 5-FU. Temperature: 42–43 °C. Time: 30 min	Surgery alone	NA	NA	NA	NA	86.8%	53.4%	NA	NA	HR: 0.20 (95% CI: 0.068–0.61, *p*: 0.005)	

NA = Not available; CHPP = Continuous Hyperthermic Peritoneal Perfusion; MMC = Mitomycin C; CDDP = cisplatin; 5-FU = Fluorouracil; UFT = Tegafur/uracil; Adj.CTx = Adjuvant chemotherapy; SA = Surgery Alone.

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
