# Peer review of "Prophylactic Hyperthermic Intraperitoneal Chemotherapy (HIPEC) for Gastric Cancer—A Systematic Review"

_jcm, 2019, doi:10.3390/jcm8101685_

Round 1

Reviewer 1 Report

Comment to Authors

This manuscript is a systematic review that investigated the usefulness of prophylactic hyperthermic intraperitoneal chemotherapy (HIPEC) for gastric cancer patients without proven metastatic disease who had undergone resection with curative intent. They extracted a total of 11 related studies and compared the overall survival, 5-year survival, disease-free survival, peritoneal recurrence, and post-operative morbidity and mortality of the patients who underwent surgery combined with pulse-prophylactic HIPEC to those who underwent surgery alone. They demonstrate that prophylactic HIPEC for gastric cancer can be performed safely, may prevent peritoneal recurrence and may prolong survival, although the studies that they selected were heterogeneous and outdated. This study was well conducted and the methods used are appropriate. The data are presented clearly. This article will be of interest to clinicians involved in the treatment of gastric cancer and may contribute to enhancing well-designed trials conducted according to current standards in the future.

However, the following minor issues require clarification:

Is it common to include ASCO abstracts and ASCO posters in a database search for systemic reviews? (P8L161) Please discuss the reasons for the conflicting results regarding the morbidity and mortality rates in previous studies.

I hope these comments will be helpful for improving this manuscript.

Reviewer 2 Report

Dear Editor,

Thank you for giving me the opportunity to review this interesting study.

This systemic review was conducted in order to summarize the data available in the literature on prophylactic HIPEC for gastric cancer patients.

The methodology used was adequate, as were the terms and databases.

The results are presented in tables 1 to 4.

I shall now comment the manuscript section by section.

Introduction is good.

Methods:

We know that the type of radical surgery (D1 vs D2) influences the prognosis of patients with gastric cancer. Was it considered in the selection of articles to include in this review? Method is good otherwise.

Results:

The tables show patients with high risk T scores (T3-T4, linitis plastica). Patients with positive clinical lymph nodes are also considered high risk patients. Why are they not included in this review? Or did all the T3-T4 patients have positive lymph nodes pre-operatively? This section is good otherwise. The important perioperative complications were reported, as for the oncological follow-up.

Discussion:

To my knowledge, it has not been demonstrated that performing HIPEC before completing the anastomoses is better than after. This is not a standard of care and should not be mentioned as such in this article. I don’t understand the rationale of performing HIPEC “several weeks” after the index surgery. Whenever we consider performing HIPEC, with no regards for the type of cancer, we usually aim to perform it before any adhesion develops, whether it is for traditional HIPEC or EPIC. Again, I don’t think this should be mentioned in this article as a proven tool or even as a good alternative.

Overall:

As mentioned by the authors, this systematic review is strongly limited. It includes only 2 randomized trials that are 30 years old and the data is very heterogenous. The primary endpoint was overall survival and this information is not available for more than half of the studies included. The type of surgery, type of HIPEC regimens and perioperative treatments were very different from one study to another. As mentioned by the authors, the regimens of perioperative chemotherapy have changed a lot over the last 30 years, which makes comparison between these studies hazardous. While I don’t think we can apply the conclusions of this article to our practice, because of the previously mentioned limitations, I think this article is still useful to demonstrate the heterogeneity of the data available on this interesting subject and the need for studies like the GASTRICHIP trial.

As for the abstract, I don’t think it is reasonable to conclude that this study “demonstrates” that prophylactic HIPEC may prevent peritoneal recurrence and prolong survival. I would rather use the terms “tends towards” or “tends to demonstrate”.

Sincerely

Reviewer 3 Report

In Europe, gastric cancer represents the fifth most common cancer. It remains diagnosed at advanced stage (serosal and/or lymph node involvement). The recommendation for curative treatment combines today perioperative systemic chemotherapy and gastrectomy with D1-D2lymph node dissection. Despite this therapeutic management, 5-year survival rates of T3 and/or N+ patients remain under 30%. More than 50% of recurrences are peritoneal and/or locoregional.

The use of adjuvant Hyperthermic Intraperitoneal Chemotherapy (HIPEC) that kills free cancer cells that can be released into peritoneal cavity during the gastrectomy and prevents peritoneal carcinomatosis recurrences, was extensively evaluated by several randomized trials conducting in Asia. One recent meta-analysis reported in your paper that adjuvant HIPEC significantly reduces the peritoneal recurrences and significantly improves the overall survival. It seems very important to validate on European or caucasian patients the results observed in trials performed in Asia.

I agree with comments in your manuscript. 

Author Response

We thank the author for taking the time reviewing our paper.